# The Role of Carcinogenesis-Related Biomarkers in the Wnt Pathway and Their Effects on Epithelial–Mesenchymal Transition (EMT) in Oral Squamous Cell Carcinoma

**DOI:** 10.3390/cancers12030555

**Published:** 2020-02-28

**Authors:** Yunpeng Bai, Jingjing Sha, Takahiro Kanno

**Affiliations:** Department of Oral and Maxillofacial Surgery, Faculty of Medicine, Shimane University, 89-1 Enya-Cho, Izumo, Shimane 693-8501, Japan; edward12@med.shimane-u.ac.jp (Y.B.); jjssxty@med.shimane-u.ac.jp (J.S.)

**Keywords:** OSCC, WNT, EMT, PMD, biomarker

## Abstract

As oral squamous cell carcinoma (OSCC) can develop from potentially malignant disorders (PMDs), it is critical to develop methods for early detection to improve the prognosis of patients. Epithelial–mesenchymal transition (EMT) plays an important role during tumor progression and metastasis. The Wnt signaling pathway is an intercellular pathway in animals that also plays a fundamental role in cell proliferation and regeneration, and in the function of many cell or tissue types. Specific components of master regulators such as epithelial cadherin (E-cadherin), Vimentin, adenomatous polyposis coli (APC), Snail, and neural cadherin (N-cadherin), which are known to control the EMT process, have also been implicated in the Wnt cascade. Here, we review recent findings on the Wnt signaling pathway and the expression mechanism. These regulators are known to play roles in EMT and tumor progression, especially in OSCC. Characterizing the mechanisms through which both EMT and the Wnt pathway play a role in these cellular pathways could increase our understanding of the tumor genesis process and may allow for the development of improved therapeutics for OSCC.

## 1. Introduction

Oral cancer is the sixth most common cancer worldwide, and oral squamous cell carcinoma (OSCC) is the predominant type [1]. Despite advances in therapy, OSCC has a relatively poor prognosis: the 5-year chance of survival is only 50% [2] due to local invasion, spread to lymph nodes, and distant metastases [3]. Aggressive treatment modalities for advanced OSCC lesions can also be severely debilitating for the patient [4]. Presumably, the identification and management of OSCC at the potentially malignant disorder (PMD) stage would improve survival rates and patient quality of life [5,6]. 

The majority of OSCC lesions develop from PMDs [7,8]. The World Health Organization (WHO) defines PMDs as the risk of malignancy being present in a lesion or condition either during the time of initial diagnosis or at a future date [9,10]. During the dynamic process of PMD development, oral epithelial dysplasia (OED) and carcinoma in situ/oral intraepithelial neoplasia (CIS/OIN) are considered criteria for identifying mild, moderate, and severe precancerous conditions [10,11]. 

Although OED and CIS are defined based on the presence of dysplastic cells in the epithelium, obtaining accurate clinical and histopathological diagnoses remains challenging [12,13]. Therefore, a more accurate and reliable system for detecting high-risk lesions is required.

Molecular or histological markers that allow for a stricter differentiation between the diagnosis of healthy tissue and PMDs are required because the histopathological diagnosis of OSCC has been established in routine paraffin-embedded specimens [14,15].

In recent studies, various biomarkers have been used to identify PMDs, including the kinase inhibitor of cyclin-dependence p16INK4a protein (p16) [16], nuclear protein ki67 [17], and the tumor suppressor p53 [18]. Additionally, functional dysregulation of genes of the wingless (Wnt) signaling pathway can promote PMD development and progression [19]. With an aberrant Wnt pathway, numerous cytokine expression patterns are affected [20]. For example, the Ca^2+^-dependent transmembrane glycoprotein E-cadherin, which is involved in cell–cell adhesion and rearrangement of epithelial cells [21], is decreased by Wnt5a dysregulation in breast cancer. The resulting decrease in cell adhesion and increase in cell motility can result in cancerogenesis [22]. Recent studies have characterized the epithelial-to-mesenchymal transition (EMT). Notably, OSCC mainly consists of epithelial dysplasia, loss of epithelial differentiation, and gaining a mesenchymal phenotype. However, further studies focusing on the EMT are required [23].

Specific carcinogenesis-related biomarkers are simultaneously involved both in the EMT and Wnt signal pathway, their synergistic action leading to PMDs, and finally OSCC [24]. Therefore, we explored how the Wnt cascade affects existing biomarkers and the precise role of these markers in EMT during the transition from normal tissue to PMDs to OSCC. 

## 2. Canonical/Non-Canonical Wnt Signaling Pathway

Wnt signaling is known to control diverse biological processes and functions. This signaling can be subdivided into two types: β-catenin-dependent signaling or canonical Wnt signaling. Another type of signaling is β-catenin-independent signaling, known as non-canonical Wnt signaling. Taking cancer stem cells (CSCs) as an example, the canonical Wnt signaling cascade was shown to be involved in the self-renewal of stem cells as well as the proliferation and differentiation of progenitor cells [25,26]. However, non-canonical Wnt signaling cascades participate in the maintenance of CSCs, directional cell movement, and inhibition of the canonical Wnt signaling cascade [27,28].

In the canonical pathway, the Wnt protein subtype binds to receptors on the cell membrane and triggers the β-catenin cluster in the cell nucleus, subsequently activating the T-cell/lymphoid-enhancing-factor transcription factors (TCF/LEF). Through this pathway, cell proliferation, apoptosis, and transformation are regulated [29]. 

In the non-canonical pathway, various receptors or co-receptors on the cell surface and extracellular matrix are involved in the pathway [30]. For non-canonical Wnt signaling, the majority of Wnt proteins bind to multiple Frizzled (Fzd) receptors, which belong to a family of G-protein-coupled transmembrane proteins [31]. Other Wnt proteins can bind to Ror1 or Ror2, RYK, and low-density lipoprotein receptor-related protein 5/6 (LRP5/6), which act as transmembrane co-receptors to form a complex with Fzd (Figure 1) [30,31,32,33].

In total, 19 Wnt genes have been identified that encode secreted cysteine-rich proteins and play a role in initiating Wnt signaling [34]. Wnt1, Wnt3, Wnt3a, Wnt7a, Wnt7b, Wnt8a, Wnt10a, and Wnt10b activate the canonical Wnt pathway [35]. For the non-canonical Wnt pathway, the best-known activator proteins are Wnt4, Wnt5a, and Wnt11 [36]. Interestingly, numerous studies have shown that specific activators, such as Wnt5a, activate both canonical and non-canonical Wnt signaling [37,38,39].

The canonical/non-canonical Wnt pathway is also regulated by several endogenous secreted Wnt inhibitors, including two types: one type binds with Wnt ligands to inhibit both canonical and non-canonical Wnt signaling, including secreted Frizzled-related protein (SFRP) family and Wnt inhibitory factor 1 (WIF1); the other type is a Wnt inhibitor that interacts with LRP5/LRP6 corepressors to block canonical Wnt signaling, e.g., the Dickkopf (Dkk) family [40]. In addition, the function of the intracellular negative Wnt regulators, such as Runt-related transcription factor 3 (RUNX3), inhibits the oncogenic Wnt signaling pathway by forming a complex with the β-catenin/TCF/LEF complex and preventing it from binding to the promoter region of target genes, such as c-Myc and cyclin D1, which regulate cell proliferation, apoptosis, and invasion (Figure 2) [41,42]. 

## 3. Epithelial Cadherin

For physical interactions between cells, a family of adhesion molecules, the cadherins, are particularly important for dynamic regulation of adhesive contacts that are associated with diverse morphogenetic processes [47]. During the embryo phase, cadherins control the separation or fusion function of tissue masses, regulate cell shapes and rearrangements [48,49,50], regulate conversions between histological cell states (e.g., epithelia versus mesenchyme), and play a role in cell long-range migration and synapse formation [51,52,53]. Even in adult tissues, cadherins play a role in and mediate various biological processes, such as the physiological regulation of epithelial and endothelial cell junctions and the maintenance of stable tissue organization to prevent the dissociation and spread of tumor cells [47,54]. Among this glycoprotein family, epithelial cadherin (E-cadherin) is the most commonly studied cadherin. E-cadherin is a calcium-dependent transmembrane glycoprotein involved in cell-–cell adhesion and rearrangement of the epithelial cell [24,47]. Downregulating E-cadherin expression plays a key role in EMT and leads to reduced cell adhesion and increased cell migration and invasion [55]. Decreased expression of E-cadherin is associated with increased invasiveness and a poor prognosis of OSCC [56,57].

Classic cadherin forms a core protein complex known as the cadherin–catenin protein complex. In the extracellular domain, the cadherin consists of five cadherin-type repeats bound together by calcium ions to form a parallel (*cis*) cadherin dimer formation [47]. In the cytoplasmic domain, the core universal-catenin complex consists of p120 catenin bound to the juxta-membrane region and β-catenin bound to the distal region, which in turn binds α-catenin. Although α-catenin binds to actin or actin-binding proteins, the mechanism remains unclear [58,59]. 

When β-catenin is associated with E-cadherin, it plays a role in establishing epithelial structure or maintaining cell–cell adhesion. After activation of canonical WNT signaling, cytoplasmic β-catenin dissociates from the cadherin-catenin protein complex, translocates, and accumulates in the nucleus, where β-catenin interacts with TCF/LEF to induce downstream gene expression (Figure 3). In this way, cell proliferation, migration, or invasion can be affected [60]. 

Jensen DH et al. [61] reported that E-cadherin is not always required during the OSCC histological multi-step process, and this has been supported by other studies. Prgomet Z et al. [24] reported that Wnt5a expression gradually increases throughout the process of carcinogenesis. Wnt5a, as an activator protein for the Wnt pathway, plays a role in both canonical and non-canonical Wnt signaling. However, Prgomet Z et al. did not find any correlation between low membranous expression of E-cadherin and high expression of Wnt5a. In addition, as proposed by Ren D et al. [62], the expression of either Wnt5a or Ror2 in Snail/A431 cells results in the inhibition of in vitro cell motility and invasiveness, without upregulation or downregulation of E-cadherin. This suggests that E-cadherin may not play a role in cell signaling but rather in cell adhesion in OSCC tissues. Thus, E-cadherin may be associated with invasiveness. However, it is not sufficiently sensitive or sluggish to reflect the rapid changes in the Wnt pathway and/or tumor. Additionally, evaluating the trend of PMDs based on cancer biomarkers in the clinical setting is very important. The aberrant expression of E-cadherin may be associated with the characteristics of cancerous tissue. Therefore, additional immunohistochemical experiments should be performed with a larger patient cohort to further explore the mechanism of E-cadherin expression. 

## 4. Vimentin

Vimentin, a 57 kDa protein, is one of the most widely expressed and highly conserved proteins of the type III intermediate filament (IF) protein family, which includes three top non-muscle cell cytoskeletal proteins. Vimentin is ubiquitously expressed in normal mesenchymal cells and is known to maintain cellular integrity and provide resistance against stress. Increased Vimentin expression has been reported in various epithelial cancers [63]. Using cryo-electron microscopy and cryo-electron tomography 3D reconstruction, Goldie KN et al. [64] showed that Vimentin contains a highly conserved α-helical “rod” domain that is flanked by non-α-helical N- and C-terminal end domains, termed the “head (77-residue)” and “tail (61-residue)”, respectively. In addition, common features shared among members of the IF family are attributed to the presence of a coiled-coiled α-helical domain that plays a role in the formation of highly stable polymers, the stability of which is controlled by the phosphorylation status of integral proteins [63]. The head domains of a Vimentin dimer are known to form symmetrical structures and three sites have been identified that show an increase in the distance between the head regions upon phosphorylation, a post-translational modification that regulates Vimentin assembly/disassembly [63,64,65].

Vimentin is encoded by a single-copy gene located on chromosome 10p13. Recently, several *cis*-elements and associated factors have been identified within the human Vimentin promoter, suggesting that the Vimentin gene is subjected to complex control [66]. One of the Vimentin promoters is believed to be a target of the β-catenin/TCF/LEF transcription factor, binding to the putative 468 bp site upstream of the transcription initiation site of Vimentin, suggesting involvement in the invasion or migration of epithelial cells [63]. In vitro research by Gilles C et al. [67] supports this theory. In their Vimentin-expressing cells, they detected the preferential cytoplasmic and nuclear localization of β-catenin, suggesting that Vimentin-expressing cells have stronger β-catenin/TCF/LEF transcriptional activity than Vimentin-negative cells. They also found that the human Vimentin promoter was upregulated by β-catenin/TCF/LEF cotransfection, and that mutation of the putative β-catenin/TCF/LEF binding site resulted in a diminution of this upregulation. 

As discussed above, β-catenin plays an important role in the canonical Wnt pathway. Upon activation of the Wnt signaling pathway, β-catenin is disassociated from the E-cadherin complex and translocated to the cytoplasm and nucleus. This suggests that the Wnt pathway triggers the transition of β-catenin from juxta-membrane to the core, subsequently activating the TCF/LEF transcription factors that upregulate the Vimentin promoter. Qiao B et al. [68] used miRNA-27a-3p (miR-27a-3p) to transfect oral squamous carcinoma stem cells (OSCSCs) and found that while miR-27a-3p was upregulated, secreted Frizzled-related protein-1 (SFRP1) and E-cadherin expression decreased. Nevertheless, the Wnt/β-catenin signaling pathway-related proteins upregulated Vimentin. SFRP1 is a known target gene for the miR-27a family at the transcriptional level; it is an antagonist of the Wnt signaling pathway and binds to Wnt proteins through its cysteine-rich domain in a competitive manner against the transmembrane Fzd receptor, resulting in inhibition of the Wnt signaling pathway [69,70,71] (Figure 4). 

Some recent studies have reported that SFRP1 either promotes or suppresses Wnt/β-catenin signaling based on the cellular context, concentration, and the expression pattern of Fzd receptors [70,72]. However, Qiao B et al. found that the expression of SFRP1 in OSCSCs remained low after the transfection of miR-27a-3p compared with other control groups, leading to the upregulation of the Wnt/β-catenin signaling pathway of Vimentin. This finding also supports a positive correlation between the canonical Wnt pathway and Vimentin expression. However, few studies have explored the relationship between the non-canonical Wnt pathway and the expression of Vimentin. Dissanayake SK et al. [73] noted that Wnt5a as an effector of the non-canonical signaling pathway increased the expression of Vimentin via activation of protein kinase C without increasing β-catenin expression or nuclear translocation. Wnt5a is a dual activator of both the canonical and non-canonical Wnt signaling pathway [37], but the effect of the canonical Wnt pathway on the expression of Vimentin remains unclear. The mechanism of how both canonical and non-canonical Wnt pathways regulate the expression of Vimentin requires further study.

## 5. Adenomatous Polyposis Coli

Adenomatous polyposis coli (APC) is a tumor suppressor gene; it was cloned from 5q21-22 as a causal gene in familial adenomatous polyposis [74]. This gene is composed of three 15-aa tandem sequences and seven 20-aa tandem sequences that bind to β-catenin and glycogen synthase kinase (GSK)-3β, respectively [75,76]. There are also three Axin-binding extension structures located in this region, known as Ser-Ala-Met-Pro (SAMP) repeats [77]. In Wnt signaling, the “destruction complex” plays an important role in the regulation of β-catenin stability; it consists of APC, Axin, GSK-3β, β-catenin, and casein kinase 1 (CK1) (Figure 2) [76]. APC protein localizes to the cytoplasm but is regulated by nuclear localization signal (NLS) and nuclear export signal (NES)—APC shuttles between the cytoplasm and nucleus [78,79,80,81,82]. A previous study found that HT-29 cell–cell contact was disrupted within 16 h by harvesting cells with a trypsin/ethylenediaminetetraacetic acid (EDTA) treatment, where APC is translocated from the cytoplasm to the nucleus. However, five days after plating cells, APC relocated to the cytoplasm [83]. According to Neufeld KL et al. [81], this type of motion pattern or nuclear accumulation of APC protein could affect the Wnt signaling pathway. Nuclear APC directly affects the abundance or activity of nuclear β-catenin through binding and destruction, so it plays an important role in PMDs [84,85]. A recent study reported that APC5 overexpression in SW480 CRC cells downregulates β-catenin and c-Myc expression [86]. Once APC is mutated, it no longer binds to β-catenin, resulting in an accumulation of β-catenin in the nucleus without degradation. Consequently, the Wnt cascade is positively regulated. This phenomenon has been detected in colorectal cancer; there are mutations in approximately 80% of the APC genes [87]. Mutations in APC have also been reported in OSCC [88,89]. Tsuchiya R et al. [85] found that the expression of APC protein was significantly increased in well-differentiated OSCCs when compared with normal tissue, and moderate and poorly differentiated SCCs in an immunohistological analysis of frozen section slides from 23 patients. The APC transcriptional product represents the differentiation of cancer cells. When epithelial dysplasia is strengthened, APC genes are strongly expressed, which was supported by the findings of Fagman H et al. [90]. APC expression is mainly cytoplasmic in cells that exit the cell cycle after serum starvation or at a high cell density, whereas APC accumulates in the nucleus in high Ki-67 expression cells. APC is known to localize from the cytoplasm to the nucleus with an increased degree of differentiation [81,85] (Figure 5). Overexpression of the APC gene induces cell cycle arrest at G0/G1-S and G2/M phases, inhibiting cell proliferation [91,92]. Xu M et al. [86] reported that APC5 may induce G1/S arrest in SW480 cells and that the synergistic effect with Axin could enhance cell growth arrest. Therefore, APC as a negative effector of Wnt signaling and cell proliferation plays a role in cell activation regulation. Notably, overexpression of APC in the nucleus may be indicative of aberrant cell differentiation. 

## 6. Snail

Snail was first identified in Drosophila, and its homologs have subsequently been found in many other species, including humans [93]. It is a member of the zinc finger family of transcriptional factors, and its activity is an important determinant of the EMT in the contexts of both mesoderm development and tumor progression [93,94,95]. A recent study found that wild-type Snail (Snail-WT) is very unstable, with a half-life of about 25 min, especially with excess expression of β-transducin repeat-containing protein (β-TrCP) [96]. β-TrCP may interact with Snail-WT through a specific phosphorylation motif that is required for binding of β-TrCP with substrate [96]. As endogenous Snail is barely detectable, the mRNA level is commonly used to monitor expression [97].

A recent study found that a wide range of signaling pathways induce Snail expression, including TGF-β [98,99,100], Notch [101], and Wnt pathways [102,103]. In this review, we explore the Wnt signaling pathway and provide a focused discussion on the molecular mechanism of the interaction between the Wnt signaling cascade and Snail expression. 

Yang Y et al. [104] found that inhibition of Connexin 32 (Cx32) upregulates Snail expression via activation of Wnt/β-catenin signaling in Hepatocellular carcinoma (HCC). Moreover, with decreased expression of Cx32 and a reduction in E-cadherin, higher Snail expression and nuclear accumulation of β-catenin was observed in HCC tissues. Excluding Cx32, one of the most efficient regulators of Snail is GSK-3β. Many signaling pathways mediated by growth factors regulate the function of Snail via GSK-3β, which is also involved in the regulation of the Wnt pathway [97,103,104]. 

Zhou BP et al. [105] found that Snail expression decreased due to constitutive GSK-3β activity, which has also been observed in previous reports. In the ubiquitin-proteasome pathway, GSK-3β phosphorylation and the proteasome pathways are believed to be involved in the regulation of Snail [106]. According to Zhou BP et al., Snail (including two GSK-3β phosphorylation motifs separated by two proline residues) influences Snail protein stability and localization in tumor cells [105]. However, GSK-3β phosphorylates several nuclear transcription factors from the C-terminus of the molecule to the N-terminus [96,107,108]. For the regulation of Snail expression, GSK-3β binds and phosphorylates Snail (at motif 2) and induces its nuclear export. Subsequent phosphorylation by GSK-3β (at motif 1) results in the association of Snail with β-TrCP, leading to the degradation of Snail in the cytoplasm. Therefore, once these two regions are mutated, the phosphorylation and stabilization of Snail are disrupted; consequently, the metastatic potential of cancer cells could be enhanced [104,107]. 

GSK-3β is considered an essential component of Wnt cascade regulation. In an unstimulated cell, cytoplasmic β-catenin is maintained at a low level via degradation after binding to the destruction complex, which is composed of scaffolding proteins Axin, conductin, GSK-3β, and APC. In normal cytoplasm, β-catenin is targeted for ubiquitination and degradation in the 26 S proteasome by paired phosphorylation through the serine/threonine kinases CK1 and GSK-3β bound to a scaffolding complex of Axin and APC. A multiprotein complex containing the scaffold protein plays a role in the phosphorylation of β-catenin, leading to formation of a β-catenin homodimer or heterodimer with the related protein. During this process, β-catenin is phosphorylated by CK1 at Ser 45; this in turn primes GSK-3β to phosphorylate serine/threonine residues Ser 33, Ser 37, and Thr 41. Phosphorylation of the last two residues triggers ubiquitylation of β-catenin by β-TrCP and degradation in proteasomes [109]. However, in the presence of Wnt ligands, Wnt binds to the Fzd receptor, which inactivates GSK-3β in the destruction complex. Additionally, in the Wnt pathway, LRP5/6 is activated and interacts with the Fzd receptor, possibly binding and inhibiting Axin and thus inhibiting GSK-3β activity. Dishevelled (Dsh) is involved in this inactivation process and inactivates a large protein complex, including APC and CK1. Dsh blocks the degradation of β-catenin, induces β-catenin stabilization, enters the nucleus, and associates with TCF/LEF transcription factors, resulting in the transcription of Wnt target genes [33,34,104,110]. 

Therefore, the upregulation of Snail protein expression is accompanied by the accumulation of β-catenin in the nucleus, reduction of E-cadherin, and increased expression of Vimentin, which is suggestive of EMT (Figure 6). In OSCC cells, the enhanced expression of Snail is indicative of increased cell invasion and migration [111]. Wang SH et al. [112] found that the upregulated Snail and activated Wnt pathway could promote lymphangiogenesis, and that a lymphangiogenic factor plays a critical role in PMDs and cervical lymph node metastasis.

## 7. Neural Cadherin

Neural cadherin (N-cadherin) is typically expressed in tissues derived from the mesoderm and neuroectoderm, which differs from E-cadherin that originates from normal squamous epithelium. This cadherin is widely expressed in the nervous system and is associated with small adherens-type junctions at synapses [113,114,115,116]. Moreover, compared with the functions of E-cadherin on the phenotype of epithelium, N-cadherin has contrary effects: it is associated with mesenchymal cell phenotype and increasing motility and invasion [117]. Several previous studies have explored the relationship between the expression of E-cadherin and N-cadherin. Specifically, among certain types of cells such as prostate [118,119], nasopharyngeal [120], breast [121], and ovarian cells [122], there is a suppression of E-cadherin and overexpression of N-cadherin, a phenomenon called E-cadherin/N-cadherin “switching” (EN-switch, Figure 7). This results in epithelial cells expressing N-cadherin, which could increase motility and invasiveness [123,124]. Inhibition of E-cadherin by LEF1 and the persistence of N-cadherin allows cells to undergo EMT, thus promoting the migratory and invasive properties of cancer cells [125]. However, according to previous studies exploring the induction of morphological changes, an EN-switch is not required [126]. 

In head and neck SCC, some studies have found that N-cadherin expression is limited in OSCC and is not correlated with the downregulation of E-cadherin expression [127]. DI Domenico M et al. [128] found in OSCCs that the expression of N-cadherin is associated with nuclear reactivity, especially in dedifferentiated variants characterized by a poor prognosis. Additionally, previous research has revealed significant correlations between N-cadherin expression and the disease stage and differentiation degree, especially in cases with N-cadherin overexpression (>61% cases). These features are predictors of a poor prognosis and a greater tendency to relapse or metastasize [128,129]. At this time, N-cadherin expression and its role in tumor progression remain unclear; previous studies of OSCC have reported rates of N-cadherin expression ranging from 37%–52.4% and the findings support the idea that the EN-switch is involved in the progression of these carcinomas [128,129,130,131]. 

Some studies have observed that the cytoplasm of β-catenin is positively associated with N-cadherin [120,125]. Moreover, the upregulation of N-cadherin has been strongly correlated with tumor differentiation, tumor size, lymph node metastasis, and poor survival in patients with nasopharyngeal carcinoma [132]. Sun H et al. [120] found that both increased protein expression of N-cadherin and β-catenin was positively correlated with lymph node metastasis and a poor prognosis.

According to Hulin-Curtis S et al. [133], cultured human vascular smooth muscle cells infected with an adenovirus encoding dominant-negative (dn)-N-cadherin via the T cell factor promoter specifically expressed dn-N cadherin in response to activation of the Wnt/β-catenin/TCF/LEF pathway. The expression of dn-N-cadherin could be induced by LiCl by inhibiting GSK-3β and activating β-catenin. This synergistic effect between N-cadherin and β-catenin can lead to PMDs, and this finding was supported by the results of Sun H et al. [120]. Due to the unique function and role of N-cadherin in cancer, N-cadherin could be a useful marker for the diagnosis of PMDs. N-cadherin is also a potential molecular therapeutic target for the treatment of OSCCs and other types of cancers.

## 8. Discussion and Conclusions

At the stage of PMDs, tumor cells invade the surrounding stromal tissues and disseminate from the solid tumor mass. As epithelial tumor cells are typically tightly associated with their neighboring cells via E-cadherin-containing adhesion junctions, tumor cells must break these extracellular junctions before they can move out as a group or single cell and invade stromal tissues [134]. To gain this ability, epithelial tumor cells may undergo a process termed EMT, whereby epithelial cells lose apical–basal polarity and cell–cell contacts and gain mesenchymal phenotypes with increased migratory and invasive capabilities. A hallmark of EMT is the functional loss of E-cadherin and overexpression of mesenchymal markers such as N-cadherin and Vimentin. 

EMT may be induced by the secretion of specific factors including vascular endothelial growth factor (VEGF), interleukin (IL), matrix metalloprotease (MMP), tumor necrosis factor α (TNFα), and TWIST [135,136,137]. Previous research has also demonstrated that several pathways are implicated in regulation of the EMT procedure, such as TGFβ signaling, Notch signaling, and Wnt signaling [138]. Among these oncogenic pathways, GSK-3β plays an important role in cancer cell invasion and metastasis, as well as the determination of cell fate and morphology [107]. Moreover, GSK-3β phosphorylates several nuclear transcription factors such as c-Myc, p53, and Snail. The phosphorylation caused by GSK-3β could result in the degradation of Snail and the inhibition of EMT [96]. Many upstream signaling pathways regulate the function of the Snail super family (including subfamilies Snail and Slug [139]) by modulating the activity of GSK-3β. Excluding the regulation of Snail, GSK-3β also controls the expression of β-catenin in the cytoplasm. The inhibition of GSK-3β through numerous pathways, such as the Wnt cascade, results in the stabilization of β-catenin and Snail, which induces cell migration, proliferation, and oncogenesis. Therefore, as a central regulator of epithelial structure and function and a suppressor of EMT, GSK-3β may be a molecular therapeutic target.

At this time, EMT has been well-studied using in vitro cell line models; however, direct observations of EMT in vivo or in clinical specimens are limited. Previous reports have demonstrated an association of EMT with invasiveness in different cancer types [140,141] but identifying EMT-transitioning cells in vivo is complicated due to the spatial and temporal heterogeneity of EMT in human cancer and lack of reliable EMT markers that can distinguish tumor cells having undergone EMT from surrounding stromal cells [142]. Another explanation for this phenomenon is that cancer cells undergo a transient EMT stage and revert to their epithelial state through the mesenchymal-to-epithelial transition (MET) after they disseminate into distant organs or tissues [143]. Numerous EMT-inducing transcription factors inhibit proliferation and result in tumor cell growth arrest [144,145,146]. Therefore, the tumor cell must revert to the epithelial state by MET to allow for metastatic growth in distant organs. The mechanisms underlying the induction of MET remain unclear, but the withdrawal of EMT inducers may result from changes in stromal cues by the surrounding environment [143]. 

OSCC is the sixth most common cancer worldwide. Advanced OSCC, with a high lymph node metastasis rate, leads to an unfavorable prognosis and high relapse rate. Although numerous studies have suggested that certain biomarkers can be used for predicting the prognosis of cancer. Almangush A et al. has reported that most studies have focused on only a few well-studied biomarkers, such as p53, Ki67, and p16 [147]. There have been few studies using promising new biomarkers, such as E-cadherin, investigated by Sgaramella N et al. in Swedish patients [148], or the Snail reported by Zheng M et al. [149], and these studies were performed in small cohorts, leading to difficulty in utilizing these molecular biomarkers for detecting cancer in the early stages in daily practice. Early detection and treatment of cancer are the most important clinical goals. According to the WHO histopathological classification, OED or CIS/OIN are considered criteria for identifying mild, moderate, or severe pre-cancerous conditions. However, this classification system does not provide a definitive diagnosis of a precancerous condition and does not provide information on patient therapy. Previous studies have shown that the presence of EMT is a predictor of OSCC progression and a prognostic factor and could be detected during the PMD phase [150,151]. Therefore, epithelial and/or mesenchymal markers require further investigation to explore their role in early tumor onset. This may allow us to define the precise stage of tumorigenesis and will increase our understanding of the conserved functions of EMT in embryonic development and tumor metastasis.

## Figures and Tables

**Figure 1 cancers-12-00555-f001:**
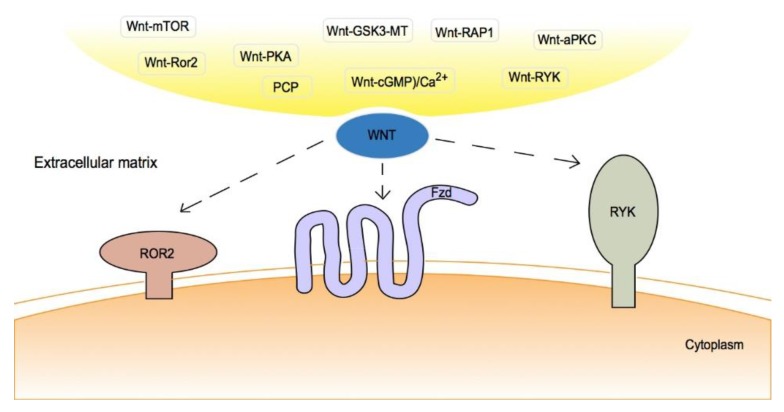
Non-canonical Wnt pathway: Wnt signaling activates another mechanism independent of β-catenin. Based on the downstream molecules activated, the non-canonical Wnt signaling can be further classified into several categories: 1. Wnt-planar cell polarity (PCP) signaling; 2. Wnt-cyclic guanosine monophosphate (cGMP)/calcium (Ca^2+^) signaling; 3. Wnt-small GTPase (RAP1) signaling; 4. Wnt-receptor tyrosine kinase-like orphan receptor 2 (ROR2) signaling; 5. Wnt-protein kinase A (PKA) signaling; 6. Wnt-glycogen synthase kinase 3 (GSK3)-microtubule (MT) signaling. 7. Wnt-atypical protein kinase C (aPKC) signaling; 8. Wnt-receptor-like tyrosine kinase (RYK) signaling; and 9. Wnt-mammalian target of rapamycin (mTOR) signaling [30,31,32,33].

**Figure 2 cancers-12-00555-f002:**
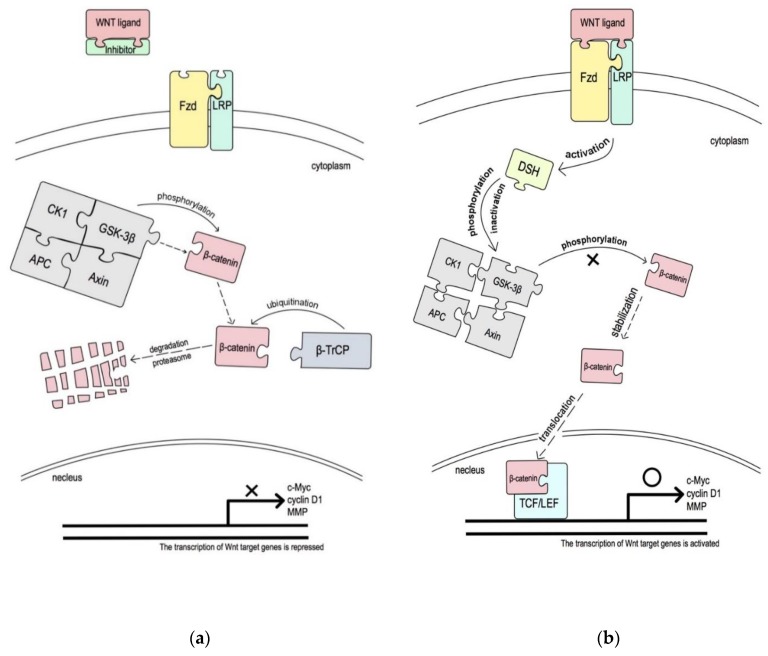
The inactivated Wnt pathway and the activated canonical Wnt pathway. (**a**) Inactivated Wnt pathway. In the absence of the Wnt ligand, cytoplasmic β-catenin is phosphorylated by glycogen synthase kinase 3β (GSK-3β) within a destruction complex that includes Axin, adenomatous polyposis coli (APC) protein, and casein kinase 1 (CK1). Phosphorylated β-catenin is directly ubiquitinated by the β-transducin repeat-containing protein (β-TrCP) E3 ligase and degraded by the 26 S proteasome pathway. Thus, the Wnt pathway is repressed due to a lack of β-catenin translocation into the nucleus [43]. (**b**) Activated canonical Wnt pathway. The Wnt ligand binds to its Frizzled (Fzd) receptor and lipoprotein-related protein (LRP) coreceptor, resulting in activation of the Disheveled (DSH) protein, which in turn destabilizes the destruction complex and leads to β-catenin stabilization via dephosphorylation. The stabilized β-catenin accumulates in the cytoplasm and translocates into the nucleus, where it interacts with the T-cell factor/lymphocyte enhancer factor (TCF/LEF) of the transcriptional complex to induce the expression of downstream genes, such as c-Myc, cyclin D1, matrix metalloproteinase 1 (MMP-1), and MMP-7 [44,45,46].

**Figure 3 cancers-12-00555-f003:**
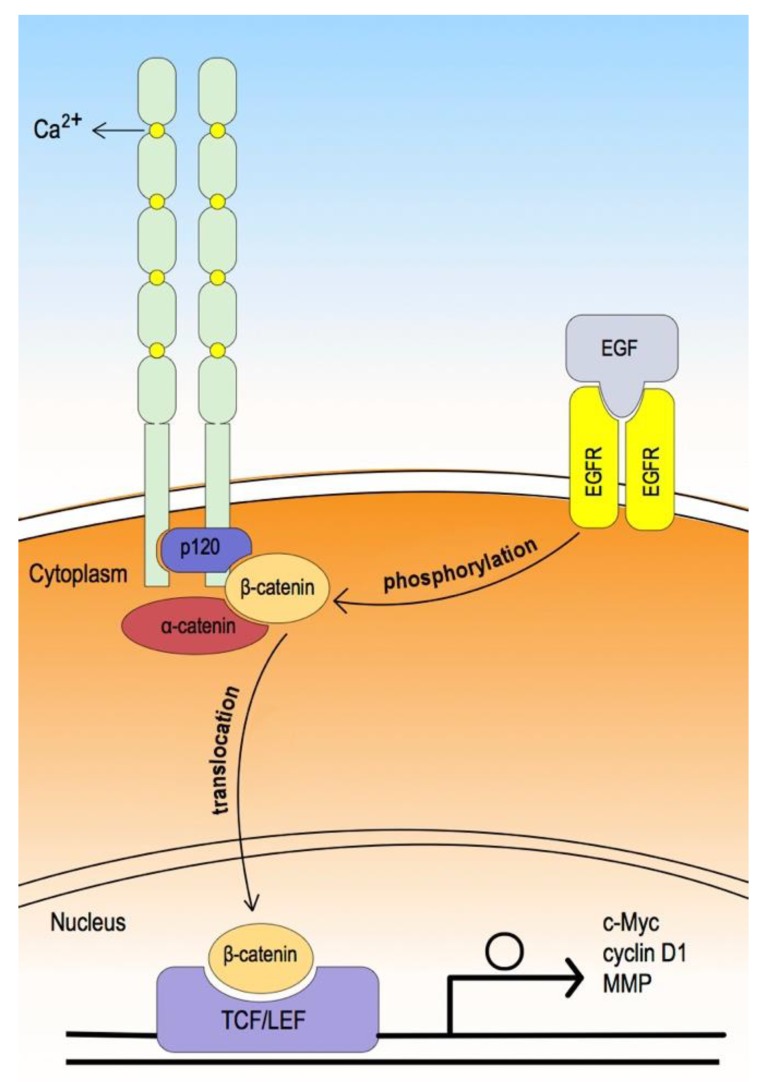
The extracellular domain consists of five cadherin-type repeats bound together by calcium ions, forming a parallel cadherin dimer. In the cytoplasmic domain, the core universal-catenin complex consists of p120 catenin bound to the juxta-membrane region, and β-catenin, bound to the distal region, which in turn binds α-catenin; α-catenin then binds through actin or actin-binding proteins [55,56]. In contrast, while epidermal growth factor (EGF) binds to its receptor (EGFR) and activates EGFR, the phosphorylation of β-catenin and GSK-3β are induced. This leads to dissociation of β-catenin from the cadherin-catenin protein complex and translocation into the nucleus (phosphorylated GSK-3β also induces uncoupling of β-catenin from the destruction complex and translocation into the core through the canonical Wnt signaling pathway). Consequently, nuclear β-catenin binds with TCF/LEF and induces the transcription of the Wnt target gene, which affects the cell cycle, migration, and invasion [56,57].

**Figure 4 cancers-12-00555-f004:**
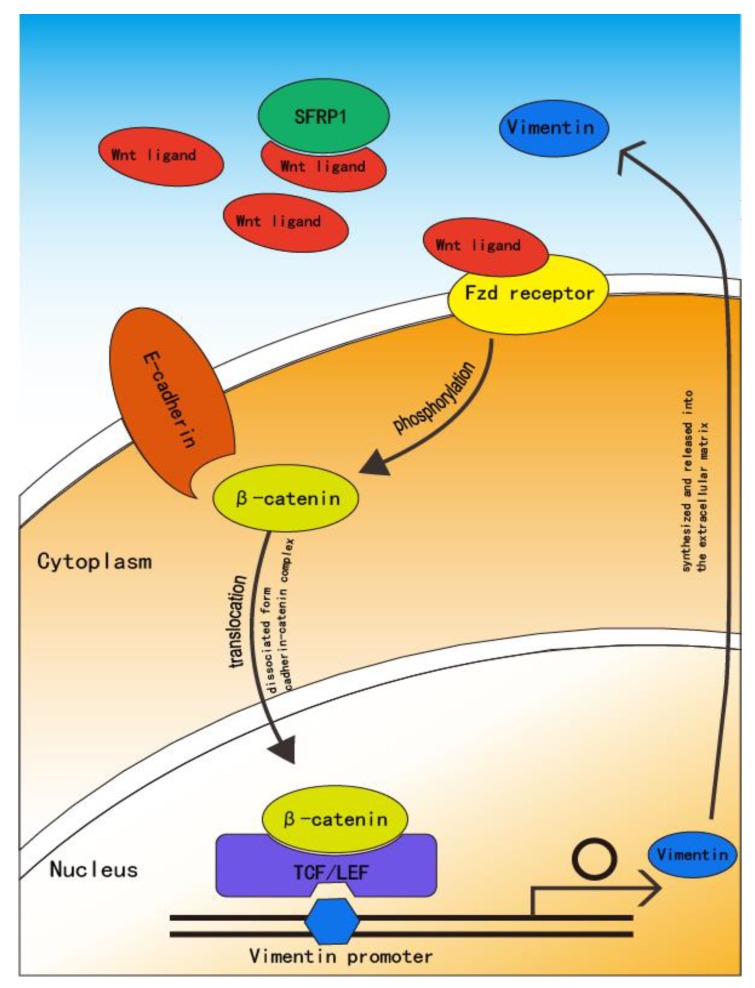
Secreted Frizzled-related protein-1 (SFRP1) is a suppressor of Wnt protein that binds to Wnt protein in a competitive manner that prevents binding with the transmembrane Fzd receptor. With decreased SFRP1 expression, the Wnt protein associates with the Fzd receptor, after which the Wnt pathway is activated. The active Wnt pathway induces β-catenin destruction through the E-cadherin complex and translocation to the nucleus where it can interact with TCF/LEF and activate the Vimentin promoter. Vimentin is then synthesized and released into the extracellular matrix [64,65].

**Figure 5 cancers-12-00555-f005:**
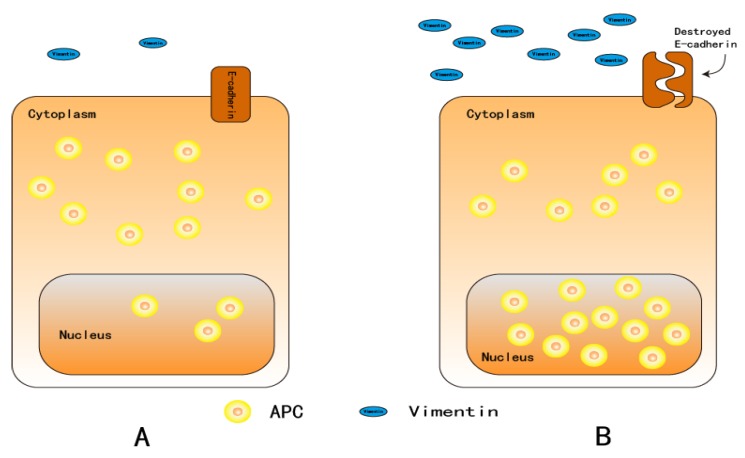
In cells where exit from the cell cycle is induced by serum starvation or high cell density, APC is mainly expressed in the cytoplasm. APC accumulates in the nucleus in cells with high Ki-67 expression. APC is known to translocate from the cytoplasm into the nucleus in cells with an increased degree of differentiation [81,85]. Nuclear accumulation of APC also occurs simultaneously with high Vimentin expression and destruction of the E-cadherin complex. **A**: Epithelial cells with a lower cell proliferation rate. **B**: Epithelial cells with a higher proliferation rate.

**Figure 6 cancers-12-00555-f006:**
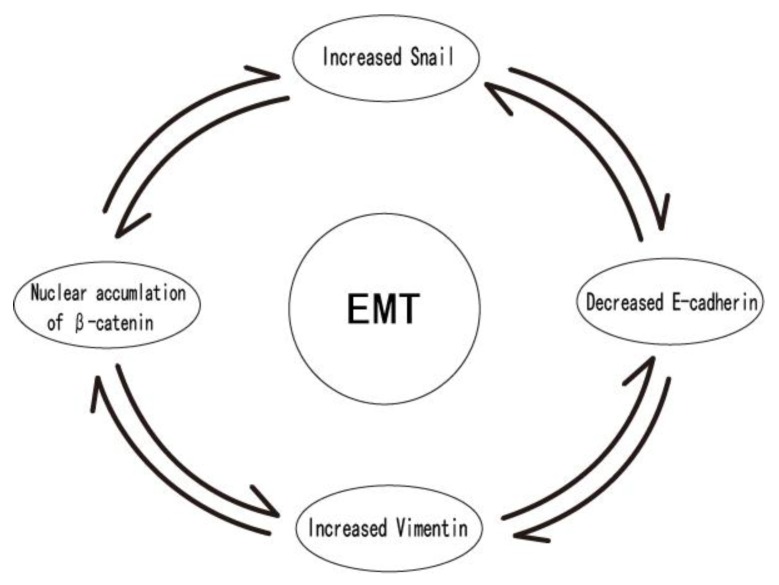
Upregulation of Snail protein expression is accompanied by the accumulation of β-catenin in the nucleus, reduction of E-cadherin expression, and increased Vimentin expression. This is suggestive of EMT. During PMD in OSCC, the enhanced expression of Snail is an indicator of increased cell invasion and migration [111].

**Figure 7 cancers-12-00555-f007:**
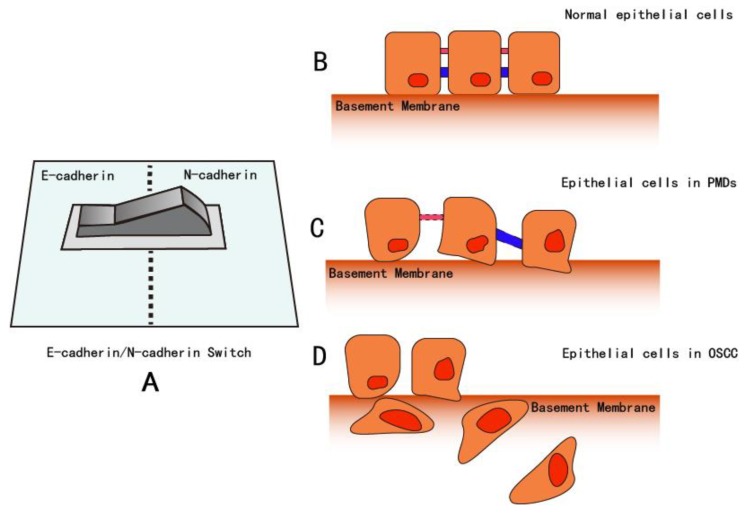
In specific cell types such as prostate [118,119], nasopharyngeal [120], breast [121], and ovarian cells [122], suppression of E-cadherin and overexpression of N-cadherin can occur, a phenomenon called E-cadherin/N-cadherin “switching”. **A**: Conceptual image of the E-cadherin/N-cadherin switch. **B**: In normal tissue, due to the intercellular bridge consisting of the desmosome, adherent proteins, such as E-cadherin, microvilli, and the epithelial cells are associated through a tight junction, forming the basement membrane. **C**: In PMDs, the intercellular junction loosens, parts of the junction and desmosome dissociate, apical-basal polarity is lost, atypical cells increase in number, and cell migration increase. In PMDs, the basement membrane becomes irregular. At the end of the PMD stage, some epithelial cells invade into deeper layers by passing through the basement membrane. **D**: In OSCCs, atypical epithelial cells increase significantly, and cell polarity is lost with nuclear expansion. At this time, the basement membrane is degraded, and cells can invade and migrate into a very deep layer. In front of the lesion, cells lose epithelial characteristics but gain mesenchymal features. In addition, mesenchymal markers such as Vimentin, Snail, and N-cadherin, are upregulated.

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
