# Peer review of "The Role of Carcinogenesis-Related Biomarkers in the Wnt Pathway and Their Effects on Epithelial–Mesenchymal Transition (EMT) in Oral Squamous Cell Carcinoma"

_cancers, 2020, doi:10.3390/cancers12030555_

Round 1
Reviewer 1 Report
The manuscript comprehensively reviews the role of various biomarkers related to the Wnt pathway and EMT in OSCC in a structured way with an excellent and exhaustive review of the literature. This review will advance the understanding of a learner regarding the Wnt pathway in progression of cancer.
Few suggestions:
-Line 63: Incorporate a brief sentence that describes the benefits of the canonical versus non-canonical pathway from a biological point of view.
-Line 194: Should read as B-catenin is dissociated….
-Line 415-416: There’s an abrupt reference to cancer of the tongue while the manuscript deals with OSCC in general and does not specify any enhanced or altered role for biomarkers in tongue cancers (which are proven to have high rate of metastasis).
Author Response
Dear reviewer 1
We are truly grateful to your critical comments and thoughtful suggestions on our manuscript. Based on these comments and suggestions, we have made careful modifications into our original manuscript. All changes made to the main text are in red and yellow-highlighted. We want to say thank you for your helpful suggestions, which were very supportive of further improving this manuscript. You will find our point-by-point responses to your comments/ questions as below.
Sincerely yours
Comments to the Author
The manuscript comprehensively reviews the role of various biomarkers related to the Wnt pathway and EMT in OSCC in a structured way with an excellent and exhaustive review of the literature. This review will advance the understanding of a learner regarding the Wnt pathway in progression of cancer.
Response: Thanks for your comments and suggestions, also, we appreciate your kind recognition of our work.
1- -Line 63: Incorporate a brief sentence that describes the benefits of the canonical versus non-canonical pathway from a biological point of view.
Response 1: Thanks for your suggestion. According to your advice, I added these sentences in the context. “Taking cancer stem cells (CSCs) as an example, the canonical Wnt signaling cascade were shown to be involved in self-renewal of stem cells as well as the proliferation and differentiation of progenitor cells [25,26]. However, non-canonical Wnt signaling cascades participate in maintenance of CSCs, directional cell movement, and inhibition of the canonical Wnt signaling cascade [27,28].” (line: 75-79).
2 - -Line 194: Should read as B-catenin is dissociated….
Response 2: Thanks for your advice, I re-wrote this sentence in line with your opinion.
3- -Line 415-416: There’s an abrupt reference to cancer of the tongue while the manuscript deals with OSCC in general and does not specify any enhanced or altered role for biomarkers in tongue cancers (which are proven to have high rate of metastasis).
Response 3: Thank you for your accurate suggestion, your question is very precise, I think other readers will also be confused with this point due to my unclear description. According to your valuable instruction, I revised this sentence as follows “OSCC is the sixth most common cancer worldwide. Advanced OSCC, with a high lymph node metastasis rate, leads to an unfavorable prognosis and high relapse rate. Although numerous studies have suggested that certain biomarkers can be used for predicting the prognosis of cancer, Almangush has reported that most studies have focused on only a few well-studied biomarkers, such as p53, Ki67, and p16 [147]. There have been few studies using promising new biomarkers, such as E-cadherin, investigated by Sgaramella in Swedish patients [148], or the Snail reported by Zheng et al. [149] and these studies were performed in small cohorts, leading to difficulty in utilizing these molecular biomarkers for detecting cancer in the early stages in daily practice.” (line: 475-482).

Reviewer 2 Report
This is a review article on the subject of Wnt signaling. It is rather complete. I have no major concerns. The authors can consider the role of porcupine, as well as the role of Rnf43.
Author Response
Dear reviewer 2
We are truly grateful to your critical comments and thoughtful suggestions on our manuscript. Based on these comments and suggestions, we have made careful modifications into our original manuscript. All changes made to the main text are in red yellow-highlighted. We want to say thank you for your helpful suggestions, which were very supportive of further improving this manuscript. You will find our point-by-point responses to your comments/ questions as below.
Sincerely yours
Comments to the Author
This is a review article on the subject of Wnt signaling. It is rather complete. I have no major concerns. The authors can consider the role of porcupine, as well as the role of Rnf43.
Response: Thanks for your comments and suggestions, also, we all really appreciate your such kind helpful commons on our work. As you mentioned, both PORCN and Rnf43 play a critical role in tumorigenesis and interact with Wnt. Especially ubiquitin ligase Rnf43, not only draw globally researchers’ attention but also my big interests. Currently, we still make an effort to explore the function of Rnf43 on both canonical and noncanonical Wnt signaling. I think in near future, concerning these two points, we will devote more time and study to get a meaningful result.
